# BACE2 distribution in major brain cell types and identification of novel substrates

Iryna Voytyuk[1,2], Stephan A Mueller[3,4], Julia Herber[3,4], An Snellinx[1,2], Dieder Moechars[5], Geert van Loo[6,7], Stefan F Lichtenthaler[3,4,8,9], Bart De Strooper[1,2,9,10]

β-Site APP-cleaving enzyme 1 (BACE1) inhibition is considered one of the most promising therapeutic strategies for Alzheimer's disease, but current BACE1 inhibitors also block BACE2. As the localization and function of BACE2 in the brain remain unknown, it is difficult to predict whether relevant side effects can be caused by off-target inhibition of BACE2 and whether it is important to generate BACE1-specific inhibitors. Here, we show that BACE2 is expressed in discrete subsets of neurons and glia throughout the adult mouse brain. We uncover four new substrates processed by BACE2 in cultured glia: vascular cell adhesion molecule 1, delta and notch-like epidermal growth factor–related receptor, fibroblast growth factor receptor 1, and plexin domain containing 2. Although these substrates were not prominently cleaved by BACE2 in healthy adult mice, proinflammatory TNF induced a drastic increase in BACE2-mediated shedding of vascular cell adhesion molecule 1 in CSF. Thus, although under steady-state conditions the effect of BACE2 cross-inhibition by BACE1-directed inhibitors is rather subtle, it is important to consider that side effects might become apparent under physiopathological conditions that induce TNF expression.

## Introduction

Currently, β-site APP-cleaving enzyme 1 (BACE1) is considered the major therapeutic target for Alzheimer's disease (AD) (Cole & Vassar, 2007; Vassar et al, 2014; Yan & Vassar, 2014; Barão et al, 2016). Together with γ-secretase, it generates amyloid-β, a short hydrophobic peptide that aggregates extracellularly into amyloid plaques in the brain of AD patients (De Strooper et al, 1998; Vassar et al, 1999; Hardy & Selkoe, 2002). Although the genetic evidence strongly supports the targeting of γ-secretases (Szaruga et al, 2017;

Voytyuk et al, 2017) to lower amyloid-β generation in the brain of patients, the severe side effects observed in phase 3 clinical trials have brought γ-secretase research almost to an end (Doody et al, 2013; De Strooper, 2014). Alternatively, BACE1 inhibition strategies have now taken center stage in the fight against the disease, and several potent, brain-penetrant compounds have entered phase 3 clinical trials (reviewed in Vassar et al, 2014; Barão et al, 2016). The development and evaluation proceed cautiously as BACE1 can cleave numerous other substrates besides amyloid precursor protein (APP) (Kuhn et al, 2012; Zhou et al, 2012; Dislich et al, 2015). These substrates have been linked to synaptic plasticity (Wiera & Mozrzymas, 2015; Hu et al, 2016; Munro et al, 2016; Pigoni et al, 2016), myelination (Hu et al, 2006; Willem et al, 2006; Fleck et al, 2013; Van Bebber et al, 2013), and axonal outgrowth (Wright et al, 2007; Hitt et al, 2012; Barão et al, 2015) among the most studied functions.

In addition, all available BACE1 inhibitors that are currently tested in the clinic cross-inhibit BACE2, the close homologue of BACE1 (Bennett et al, 2000; Farzan et al, 2000). Most of the inhibitors are equipotent (Neumann et al, 2015; Cebers et al, 2016a), with MK-8931 showing even higher selectivity for BACE2 (Kennedy et al, 2016). Similar to BACE1, BACE2 is a type I transmembrane protein that belongs to the peptidase A1 family (also called the pepsin family) of aspartyl proteases. Unlike BACE1, which is highly expressed in the brain, BACE2 is more prominently found in peripheral tissues, namely, the colon, kidney, and pancreas (Bennett et al, 2000). In pancreatic β cells, for example, BACE2 cleaves the pro-proliferative plasma membrane protein TMEM27, thereby impacting β cell mass and function (Esterházy et al, 2011). Furthermore, BACE2 suppression promotes β cell survival through another potential substrate, islet amyloid polypeptide, whose overexpression induces glucose tolerance defects (Rulifson et al, 2016; Alcarraz-Vizán et al, 2017). In this regard, inhibition of BACE2 has even been proposed for the treatment of diabetes (Esterházy et al, 2011; Stützer et al, 2013). BACE2 also processes the pigment cell–specific melanocyte protein involved in melanosome formation in pigment cells (Rochin et al,

[1]Department of Neurosciences, Katholieke Universiteit Leuven, Leuven, Belgium  [2]Centre for Brain and Disease Research, Flanders Institute for Biotechnology (VIB), Leuven, Belgium  [3]German Center for Neurodegenerative Diseases (DZNE), Munich, Germany  [4]Neuroproteomics, School of Medicine, Klinikum Rechts der Isar, Technische Universität München, Munich, Germany  [5]Discovery Neuroscience, Janssen Research and Development, Division of Janssen Pharmaceutica NV, Beerse, Belgium  [6]Center for Inflammation Research, VIB, Gent, Belgium  [7]Department of Biomedical Molecular Biology, Gent University, Gent, Belgium  [8]Institute for Advanced Study, Technische Universität München, Munich, Germany  [9]Munich Cluster for Systems Neurology, Munich, Germany  [10]Dementia Research Institute, Institute of Neurology, University College London, London, UK

Correspondence: Bart.DeStrooper@kuleuven.vib.be

2013). As a result, genetic knockout (KO) or pharmacological inhibition of BACE2 results in depigmentation, the most obvious phenotype of mouse *Bace* KO models and the most consistent side effect seen in preclinical studies of BACE1/2 inhibition (Dominguez et al, 2005; Shimshek et al, 2016; Cebers et al, 2016b; Voytyuk et al, 2017).

Thus, although some research has already indicated potential problems with peripheral inhibition of BACE2, surprisingly little is known about the function of BACE2 in the brain, the main target organ for BACE1 inhibitors. Using a sensitive and very specific in situ hybridization technique, we show here that *Bace2* mRNA is expressed in subsets of neurons, oligodendrocytes, and astrocyte-like cells lining the lateral ventricles in the mouse brain. Although all major cell types express *Bace2*, the levels vary considerably within the cell types in different regions of the brain. Using liquid chromatography coupled with tandem mass spectrometry (LC–MS/MS) to analyze the conditioned medium of cultured mouse glia, we uncover and validate four previously unknown substrates of Bace2: vascular cell adhesion molecule 1 (VCAM1), delta and notch-like epidermal growth factor–related receptor (DNER), fibroblast growth factor receptor 1 (FGFR1), and plexin domain containing 2 (PLXDC2). We find that under constitutive, nonstimulated conditions, the extent of substrate cleavage by BACE2 is low. However, we demonstrate that BACE2 cleavage of the new substrate VCAM1 gets strongly up-regulated under inflammatory conditions in vitro and in vivo. Neuroinflammation is one of the hallmarks of AD, and several inflammatory cytokines, in particular TNF and IL-1β, are known to be up-regulated in both the periphery and the brain of AD patients (Akiyama et al, 2000; Sastre et al, 2006; Wang et al, 2015; Lai et al, 2017). Our findings indicate that BACE2 may be an important secretase under inflammatory conditions in the brain.

# Results

### *Bace2* expression in the mouse brain

To characterize *Bace2* expression in the brain and compare it with that of *Bace1*, we used the RNAscope Fluorescent Multiplex Assay (ACD Bio). This in situ hybridization technique uses 20 probe pairs and four amplification steps to allow quantitative detection of a single RNA molecule with excellent sensitivity and specificity.

RNAscope in situ hybridization data show that *Bace2* is less widely expressed in the mouse brain than *Bace1*. Characterization of the expression pattern and relative abundance in different brain areas and cell types is summarized in Table 1 and examples of expression patterns are shown in Fig 1. Synaptophysin (*Syp*), glutamate aspartate transporter (*Glast*), and myelin basic protein (*Mbp*) were used as specific markers for neurons, astrocytes, and oligodendrocytes, respectively, demonstrating that *Bace2* expression can be detected in all three cell types (Fig 1A–C). It is interesting that *Bace2* expression in these cells varies across different brain regions. To evaluate the level of expression of our secretases of interest, we compared them with markers and reference genes used by RNAscope with known expression levels (Fig S1) (Zhang et al, 2014). Highest neuronal expression is found in the mouse ventral hippocampus at 4 mo (Fig 1D). The lining of the lateral ventricle is the only brain area showing *Bace2* expression in astrocytes (Fig 1E), whereas in oligodendrocytes, especially in young animals, *Bace2* is found throughout the fiber tracts (Fig 1F). Although in most cases *Bace1* is expressed in *Bace2*-positive cells (Fig 1G), the reverse is not true, as *Bace1* is much more widely expressed in the brain. For instance, *Bace1* is abundant in neuronal-rich areas, such as the hippocampus, thalamus, and cortex, showing much higher expression than *Bace2*. Many neurons, such as Purkinje cells, express only *Bace1* (Fig 1I), and there are whole neuron-rich areas, such as the dorsal hippocampus, with high *Bace1* expression, but virtually devoid of *Bace2* (Fig 1J). Some oligodendrocytes in the fiber tracts of the striatum, cerebellum, and corpus callosum express both secretases, with *Bace1* being more abundant (Fig 1H). Interestingly, expression of both *Bace1* and *Bace2* in oligodendrocytes is highest in young postnatal day (P) 16 animals, at the age of myelination onset. Few astrocytes throughout the brain focally express *Bace1*, whereas *Bace2* is only expressed at the astrocytes lining the lateral ventricle (Fig 1K).

### Identification of Bace2 substrate candidates

As *Bace2* is clearly expressed in various brain cell types, we set out to uncover its substrates. Following the approach of a previous study of BACE1 substrates in CSF (Dislich et al, 2015), we examined mouse CSF with mass spectrometry, comparing the CSF of double KO (dKO) mice with that of *Bace1*$^{-/-}$ mice (Fig S2 and Table S1). Overall, 579 proteins were quantified in at least three replicates of

**Table 1.   Summary of BACE2 and BACE1 expression in the mouse brain.**

| Brain area | BACE2 expression | BACE2 expressing cell types | BACE1 expression | BACE1 expressing cell types |
|---|---|---|---|---|
| Hippocampus (dorsal) | – | Not found | +++ | Neurons |
| Hippocampus (ventral CA3 and subiculum) | ++ | Neurons | +++ | Neurons |
| Cortex (highest in motor and somatosensory layers 4 and 5) | + | Neurons | ++ | Neurons, occasional astrocytes |
| Thalamus | + | Neurons | +++ | Neurons, occasional astrocytes |
| Cerebellum | + | Oligodendrocytes | ++ | Neurons, oligodendrocytes |
| Striatum | + | Oligodendrocytes | ++ | Oligodendrocytes |
| Fiber tracts | +, ++ at P16 | Oligodendrocytes | ++, +++ at P16 | Oligodendrocytes, astrocytes |
| Lateral ventricle lining | + | Astrocytes | – | Not found |

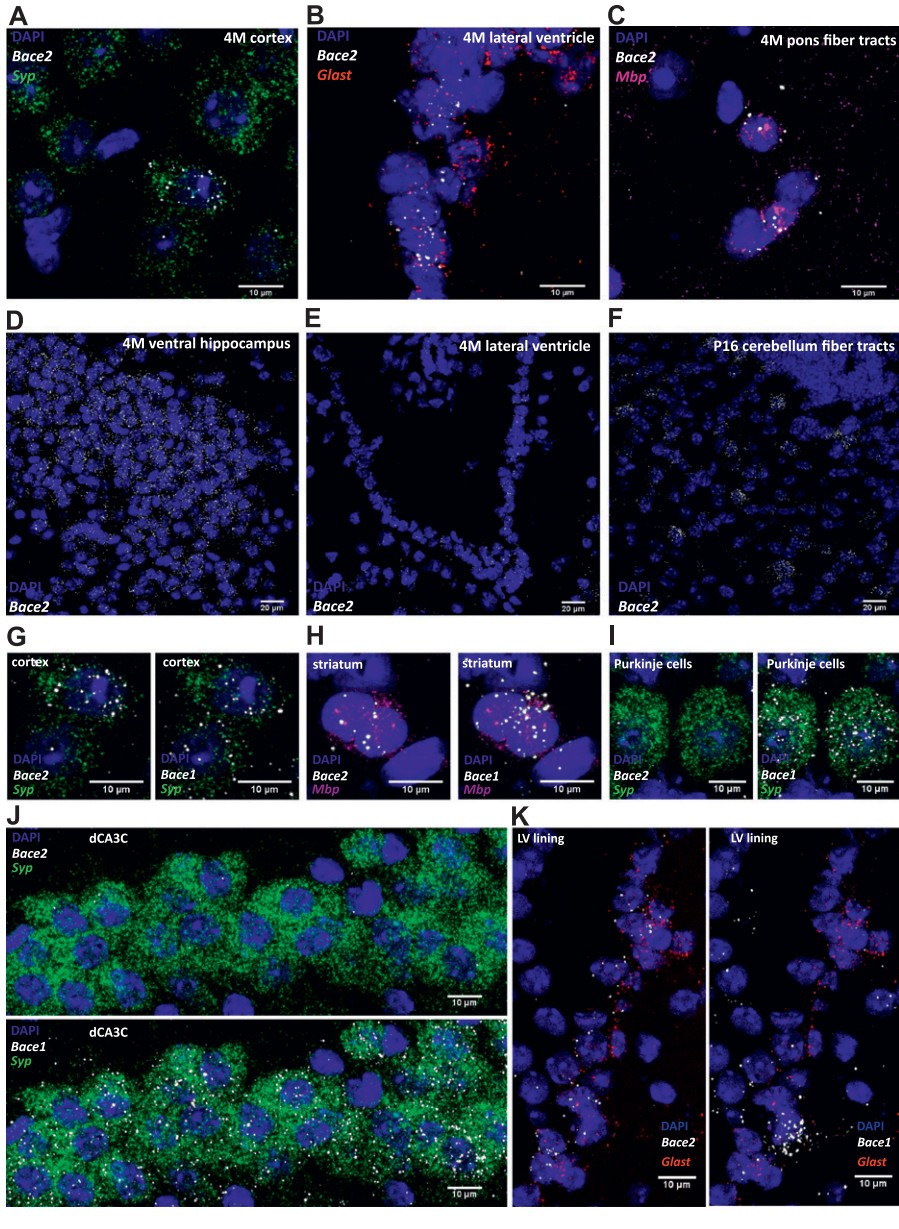

**Figure 1. *Bace2* mRNA expression in neurons, astrocytes, and oligodendrocytes.**
Examples of *Bace2* mRNA expression in neurons (A), astrocytes (B), and oligodendrocytes (C) as identified by co-localized cell type–specific markers *Syp*, *Glast*, and *Mbp*, respectively. Brain areas with highest neuronal (D), astrocyte (E), and oligodendrocyte (F) expression of *Bace2*. Neurons (G), as well as oligodendrocytes (H), expressing *Bace2* also express *Bace1*. Purkinje cells express only *Bace1* (I); similarly, dorsal hippocampus shows high *Bace1* expression, but is virtually devoid of *Bace2* (J). Few astrocytes throughout the brain focally express *Bace1*, whereas *Bace2* is only expressed by astrocytes lining the lateral ventricle (K). Representative images from two adult (4 mo) and two young (P16–20) male WT mice. *Syp*, neuronal marker; *Glast*, astrocytic marker; and *Mbp*, oligodendrocyte marker.

both, *Bace1⁻ᐟ⁻* and dKO mouse CSF. A subset of 60 proteins showed a log2-transformed ratio of at least ±0.5 (>1.41-fold or <0.71-fold) with *P*-value < 0.05 (Table S1). The analysis demonstrated pronounced increases and decreases in cytosolic proteins such as IGAC, SERPINA3M, IGHG1, IGG2AC and SCG5, LY86, OGN, and C2 (Fig S2). Among the increased proteins, we found proteins of the immunoglobulin family and a member of the serpin family of serine protease inhibitors, whose functions are not clearly understood (Silverman et al, 2001). The decreased proteins are a neuroendocrine member of the secretogranin family, a player in the innate immune response, small leucine-rich proteoglycan, and members of the complement system (Taupenot et al, 2003; Deckx et al, 2016; Hasan et al, 2017). However, as none of these proteins harbors a transmembrane domain, they are unlikely to be direct BACE2 substrates, and the observed changes are likely due to the indirect

effects of BACE2 absence (in the background of *Bace1⁻ᐟ⁻*). Single-span transmembrane proteins LRRN1, PLXDC2, CNTN2, and PTPRN2 that were previously identified as BACE1 substrates are given in bold in Fig S2 (Kuhn et al, 2012; Zhou et al, 2012; Dislich et al, 2015); they are likely direct BACE2 substrates as well, but are only partially cleaved by BACE2 (31–51% decreases reflect BACE2 contribution on top of BACE1 inhibition). This demonstrates that BACE2 inactivation alters BACE2 substrate levels in the murine CSF proteome.

Proteins in CSF may originate from all brain cell types. As shown in Fig 1, these express highly variable levels of BACE2 (Table 1). Thus, we further examined BACE2 substrate candidates with brain relevance by turning to a primary cell culture, where BACE2 was previously shown to be expressed and active (Dominguez et al, 2005). Glia were cultured in the presence or absence of the unselective inhibitor Compound J (CpJ), which blocks both BACE1 and BACE2.

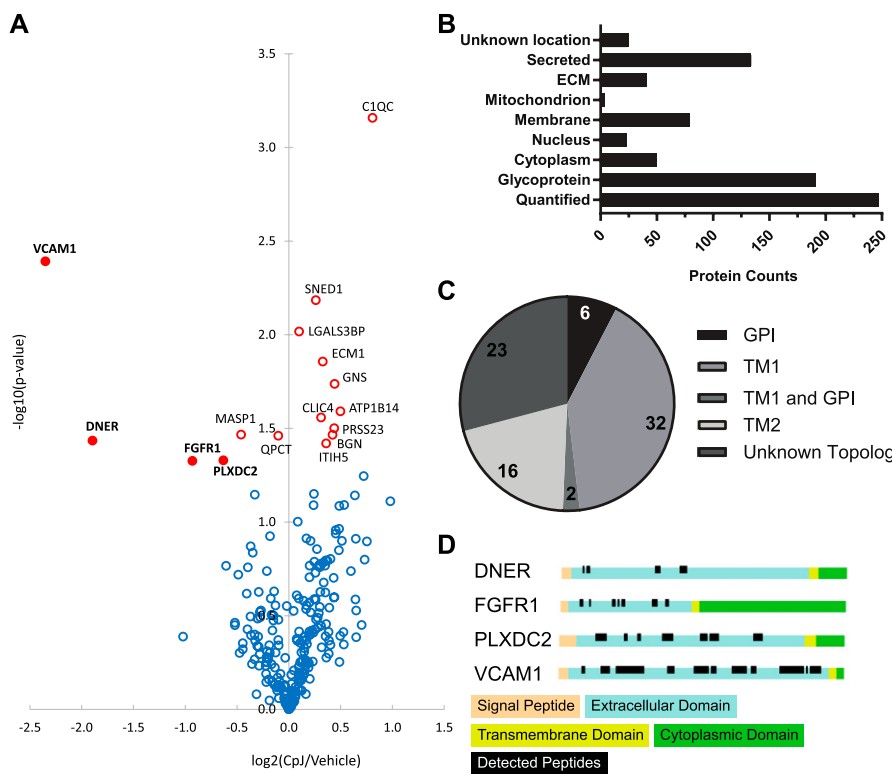

**Figure 2. BACE2 secretome in primary glia cultures.**
**(A)** Volcano plot of proteomic analysis of conditioned medium from *Bace1*$^{-/-}$ cultured glia treated with BACE inhibitor CpJ or vehicle (n = 3). For each relatively quantified protein (representing a dot on the plot), the −log10-transformed *t* test *P*-value was plotted against the log2-transformed LFQ intensity ratios of CpJ/vehicle. Proteins with a *t* test *P*-value <0.05 are marked with open red circles, whereas proteins with *P*-value >0.05 are shown in blue. Closed red dots labeled with bold letters denote the four substrate candidates that are reduced by more than 30% in the inhibitor-treated samples. No corrections for multiple hypothesis testing were applied in this discovery experiment. **(B)** Uniprot subcellular locations of the identified proteins. Glycoproteins were defined according to UniProt Keywords. **(C)** Topology of membrane proteins according to Uniprot subcellular locations. **(D)** Uniprot Topology of DNER, FGFR1, PLXDC2, and VCAM1 with mapped identified peptides.

Therefore, we decided to increase the sensitivity of our system by using *Bace1*$^{-/-}$ glia to identify proteins selectively shed by BACE2. For this purpose, we used a mass spectrometry–based screening method. Conditioned media of cultures treated with an inhibitor or vehicle only were collected after 48 h. Glycosylated membrane proteins, which constitute nearly 90% of all single-span membrane proteins, were enriched using the secretome protein enrichment with click sugars method, and proteins were quantified by using LC–MS/MS–based label-free quantification (LFQ) (Kuhn et al, 2012). Overall, 246 proteins (191 glycoproteins according to UniProt) were relatively quantified in three replicates and subjected to statistical analysis (Fig 2B and C). Six significantly decreased proteins were identified by mass spectrometry analysis in a conditioned medium of primary glia cultures treated with an inhibitor, four of which were decreased by more than 30%: VCAM1, DNER, FGFR1, and PLXDC2 (Fig 2A and Table S2). All four are, furthermore, type 1 transmembrane proteins, with large extracellular domains with several predicted glycosylation sites (Fig 2D). Only peptides matching the extracellular domains were identified (Fig 2D), which indicates that these proteins were indeed secreted. The shedding of VCAM1 and DNER was very strongly reduced by 80% and 73%, respectively, suggesting that BACE2 is the main protease cleaving these proteins, with little contribution of other "sheddases" in the glia. FGFR1 and PLXDC2 shedding is, in contrast, only moderately reduced by 48% and 35%, respectively, suggesting that other proteases are likely contributing to their cleavage.

### Validation of BACE2 substrate candidates in cell culture

The results of the mass spectrometry–based screening were further validated in cell cultures. Specific antibodies for the N-terminal part of VCAM1 and DNER are readily available, allowing their validation as substrates at the endogenous levels of expression. As shown in Fig 3A, shed VCAM1 is detected in a conditioned medium of *Bace1*$^{-/-}$ glia, and this shedding is decreased by 64% upon inhibition with CpJ (Fig 3A and B). The full-length VCAM1 shows a slight trend to accumulate in the glia cells without reaching statistical significance (Fig 3A and C). Prominent VCAM1 shedding is observed in WT and *Bace1*$^{-/-}$ glia, whereas little shedding is seen in *Bace2*$^{-/-}$ glia (Fig 3D). We next used the Bace inhibitor CpJ in *Bace1*$^{-/-}$, WT, and *Bace2*$^{-/-}$ glia to determine the effect on VCAM1 shedding. Similar to *Bace1*$^{-/-}$ glia, shedding of VCAM1 is attenuated in WT glia upon BACE inhibition (Fig 3E), whereas no change is observed in the already low level of shedding in *Bace2*$^{-/-}$ glia (Fig 3F), suggesting that VCAM1 is a selective BACE2 substrate in glia cells and not processed by BACE1.

DNER shedding into the medium is decreased by 75% upon treating the cells with CpJ (Fig 3G and H). Full-length DNER does not accumulate in the glia cells (Fig 3G and I). DNER is shed by WT glia, as well as *Bace1*$^{-/-}$ glia, but shedding is prominently decreased in *Bace2*$^{-/-}$ glia (Fig 3J). CpJ treatment dramatically decreases the shedding of DNER in WT and *Bace1*$^{-/-}$ glia (Fig 3G and K), but does not further decrease shedding of DNER in *Bace2*$^{-/-}$ glia cultures (Fig 3L).

Unfortunately, available antibodies for FGFR1 and PLXDC2 cross-react with a variety of proteins when used at the endogenous levels of expression. We, therefore, moved to a COS-1 overexpression system. Shedding of tagged murine PLXDC2 overexpressed in COS-1 cells can be monitored, and commercially available antibody raised against N-terminal FGFR1 detects the high levels of the human

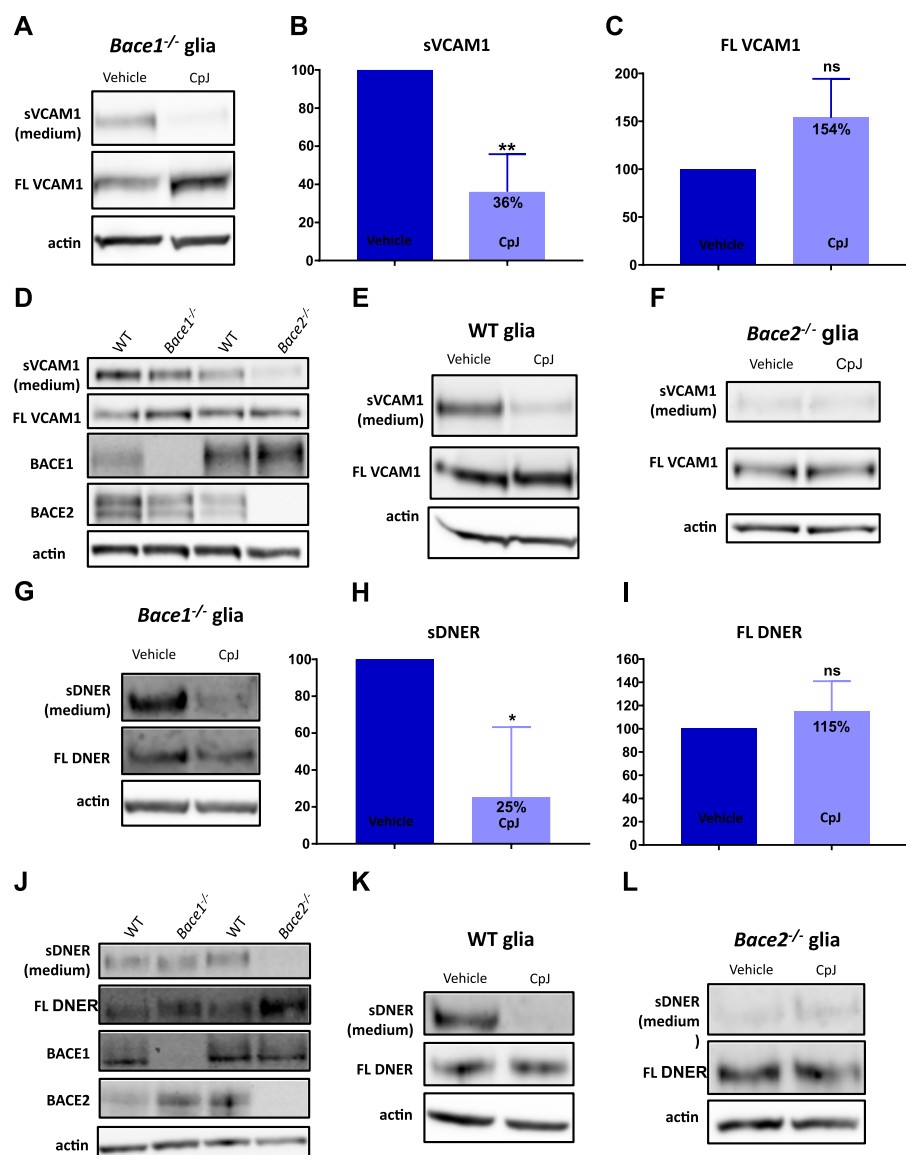

**Figure 3. Validation of VCAM1 and DNER as BACE2 substrates in primary mixed glia cultures.**
**(A)** VCAM1 in medium and cell lysates of $Bace1^{-/-}$ glia treated with vehicle or CpJ (10 µM) for 24 h (n = 4). **(B)** Quantification of VCAM1 shedding into conditioned medium of $Bace1^{-/-}$ glia treated with vehicle or CpJ for 24 h (paired $t$ test, $P$ = 0.01, n = 4). **(C)** Quantification of VCAM1 accumulation in cell lysates of $Bace1^{-/-}$ glia treated with vehicle or CpJ (10 µM) for 24 h (paired $t$ test, $P$ = 0.08, n = 4). **(D)** VCAM1 shedding in WT, $Bace1^{-/-}$, and $Bace2^{-/-}$ glia (n = 3). **(E)** VCAM1 in medium and cell lysates of WT glia treated with vehicle or CpJ (10 µM) for 24 h (n = 3). **(F)** VCAM1 in medium and cell lysates of $Bace2^{-/-}$ glia treated with vehicle or CpJ (10 µM) for 24 h (n = 3). **(G)** DNER in medium and cell lysates of $Bace1^{-/-}$ glia treated with vehicle or CpJ (10 µM) for 24 h (n = 3). **(H)** Quantification of DNER shedding into conditioned medium of $Bace1^{-/-}$ glia treated with vehicle or CpJ for 24 h (paired $t$ test, $P$ = 0.03, n = 3). **(I)** Quantification of DNER accumulation in cell lysates of $Bace1^{-/-}$ glia treated with vehicle or CpJ for 24 h (paired $t$ test, $P$ = 0.3, n = 3). **(J)** DNER shedding in WT, $Bace1^{-/-}$, and $Bace2^{-/-}$ glia (n = 3). **(K)** DNER in medium and cell lysates of WT glia treated with vehicle or CpJ (10 µM) for 24 h (n = 3). **(L)** DNER in medium and cell lysates of $Bace2^{-/-}$ glia treated with vehicle or CpJ (10 µM) for 24 h (n = 3).

overexpressed protein. Full-length FGFR1 overexpressed in COS-1 cells runs as a doublet around the 97-kD mobility marker (Fig 4A). When FGFR1 is co-expressed in COS-1 cells with human BACE1 or BACE2, a fragment between 51 and 64 kD is shed into the medium, and this is inhibited by CpJ. Thus, shedding of this fragment into the medium can be performed by both BACE2 and BACE1 in this overexpression experiment. A shed triple band running as a smear around 40 kD is only seen upon co-expression of FGFR1 with BACE2 and may represent degradation products. In fact, these bands are not seen after CpJ treatment, indicating that it is indeed BACE2 activity that is initially responsible for their production and that likely other proteases can then further cleave the shed FGFR1. FGFR1 is likely processed by additional endogenous proteases as several additional bands appear when FGFR1 is transfected alone. It is also possible that these bands are overexpression artifacts. Thus, our transfection data confirm that FGFR1 can be processed by both BACE1 and BACE2, resulting in a shed fragment with similar mobility in SDS–PAGE and various other fragments. In the absence of good antibodies, it is impossible to explore the physiological relevance of these cleavages further.

Transfected mouse PLXDC2 (N-terminally tagged with HA and C-terminally tagged with FLAG) appeared in the lysate between 64 and 97 kD as two bands, likely lower immature and upper glycosylated forms (Fig 4B), but it is this mature glycosylated form that is apparently processed by BACE1 or BACE2 as it disappears upon co-expression. Concomitantly, a shed fragment is observed above the 64-kD mark in the conditioned medium and this is increased with BACE1 or BACE2 co-expression. Shedding of PLXDC2 and disappearance of the mature form of PLXDC2 in the cell extracts are inhibited by CpJ. These overexpression data confirm that PLXDC2 can be processed by BACE2 (and BACE1) but do not yield further physiological information on the relevance of this processing event. However, this work awaits the generation of high-quality antibodies.

**A**

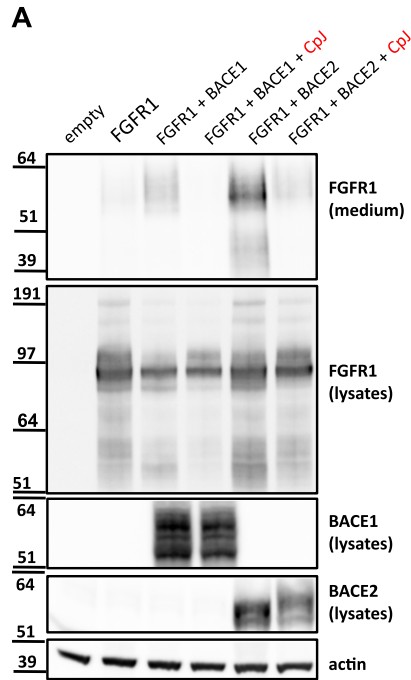

**B**

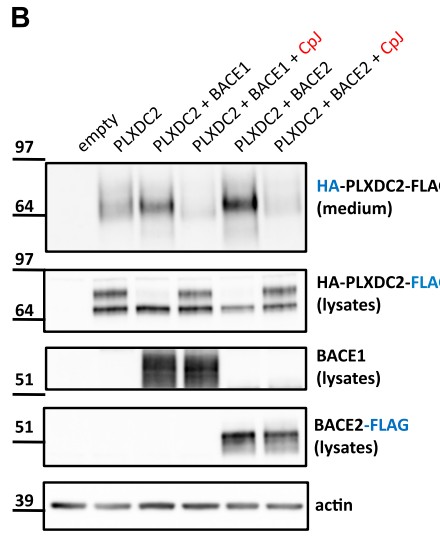

**Figure 4. Validation of FGFR1 and PLXDC2 as BACE2 substrates in COS-1 overexpression system.**
**(A)** FGFR1 in lysates and medium of COS-1 cells overexpressing FGFR1 alone (lane 2), FGFR1 with BACE1 (lanes 3 and 4), and FGFR1 with BACE2 (lanes 5 and 6). Lanes 4 and 6 were treated with inhibitor CpJ (10 μM). Lane 1 was mock-transfected with an empty vector. Representative for three experiments. **(B)** PLXDC2 in medium (top panel) and lysates (second panel from the top) of COS-1 cells overexpressing PLXDC2 alone (lane 2), PLXDC2 with BACE1 (lanes 3 and 4), and PLXDC2 with BACE2 (lanes 5 and 6). Lanes 4 and 6 were treated with inhibitor CpJ (10 μM). Lane 1 was mock-transfected with an empty vector. Control blots for BACE1, BACE2, and actin are shown for each transfection. Representative for three experiments.

## Validation of Bace2 substrate candidates in vivo

Because both VCAM1 and DNER were validated as authentic BACE2 substrates at the endogenous level in cultured glia cells, and good antibodies are available, we proceeded with their validation in CSF from WT, $Bace1^{-/-}$, $Bace2^{-/-}$, and dKO mice. The shedding of DNER and VCAM1 was compared with that of a well-known BACE1 substrate, SEZ6 (Dislich et al, 2015; Pigoni et al, 2016) (Fig 5A). As expected, the shedding of SEZ6 was significantly decreased in the CSF of $Bace1^{-/-}$ and dKO mice but not in $Bace2^{-/-}$ (Fig 5A and B), whereas no significant differences in DNER and VCAM1 were observed between genotypes (Fig 5C and D, respectively). Similarly, no differences were observed in soluble DNER in the TBS fraction of homogenized cortices from 1-year-old male mice of different genotypes (Fig S3A and B), as well as in soluble VCAM1 in TBS fractions (Fig S3C and D). The levels of full-length proteins were found to be equal as well (Fig S3E and F). Similar results were obtained using the subdissected ventral hippocampus of 4-mo-old males (Fig S3G) and P16 subventricular zone (Fig S3H), brain areas where high BACE2 expression is observed. Finally, to rule out any potential compensation mechanisms occurring in vivo in KO animals, an acute inhibition experiment was performed in 4-mo-old $Bace1^{-/-}$ males treated with CpJ by 4× i.p. injections at 12-h intervals. No changes in VCAM1 or DNER processing were observed (Fig S3I and J, respectively). Therefore, we conclude that under basal conditions, VCAM1 and DNER are not shed by BACE2 in the brain.

We reasoned that cultured glia, where we identified the four new BACE2 substrates, are not well representative of glia found in the adult mouse brain in basal conditions (Cahoy et al, 2008; Lange et al, 2012; LoVerso et al, 2015). As astrocytes become activated during injury, primary culture preparation from dissected brain

might reflect a certain degree of inflammatory phenotype (Lange et al, 2012). Interestingly, the level of one of the BACE2 substrates identified, VCAM1, is up-regulated upon inflammatory stimulus in vitro and in vivo (van Loo et al, 2006). When we added pro-inflammatory agents TNF or IL-1β to primary glia cultures, we observed the expected up-regulation of VCAM1 and, remarkably, a strong increase in VCAM1 shedding into the medium, in particular at the 24-h time point (Fig 6A). This additional shedding is performed by BACE2, as CpJ treatment of TNF-treated $Bace1^{-/-}$ glia blocked the increased shedding even when VCAM1 is up-regulated (Fig 6B). Notably, this effect of pro-inflammatory agents is specific for VCAM1 and, in contrast, leads to a down-regulation of DNER in the lysates of glia treated with TNF and IL-1β (Fig S4). Soluble DNER in the medium is already at the borderline of detection, and there are no indications of any change upon pro-inflammatory treatments (Fig S4).

Subsequently, we treated WT mice with 250 μg/kg TNF for 24 h, as was previously described to induce neuroinflammation (Biesmans et al, 2015). Shed VCAM1 was increased 2.2-fold =in the CSF of mice treated with TNF (Fig 6C and D). Surprisingly, we did not observe any increase in VCAM1 in either the TBS fraction or total cell lysates of brain homogenates (Fig 6C). We finally compared the responses to TNF in WT, $Bace1^{-/-}$, and $Bace2^{-/-}$ mice and found that both WT and $Bace1^{-/-}$ mice show up-regulated VCAM1 shedding into CSF upon 24-h treatment with TNF, whereas VCAM1 shedding in $Bace2^{-/-}$ mice is not affected (Fig 6E and F). Thus, BACE2 is responsible for the additional shedding of VCAM1 in pro-inflammatory conditions in vitro and in vivo.

## Discussion

The role of BACE2 in the pancreas and skin is generally recognized (Esterházy et al, 2011; Rochin et al, 2013; Shimshek et al, 2016), but

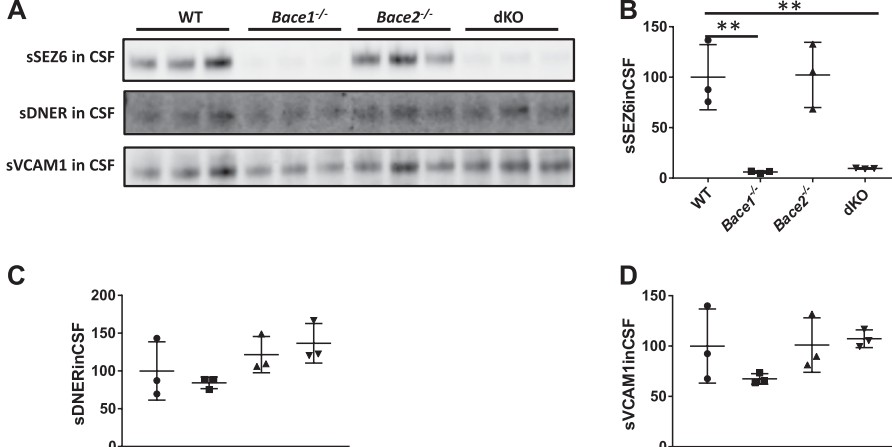

**Figure 5. Absence of effects on VCAM1 and DNER in CSF of $Bace^{-/-}$ mice.**
**(A)** Shed VCAM1 and DNER in triplicates of CSF of 11-mo-old WT, $Bace1^{-/-}$, $Bace2^{-/-}$, and dKO male mice, as compared with a known BACE1 substrate—SEZ6. **(B)** SEZ6 is significantly reduced in $Bace1^{-/-}$ and dKO, but not $Bace2^{-/-}$ CSF compared with control WT CSF (one-way ANOVA: $P = 0.006$, $P = 0.008$, and $P = 1.00$, respectively; $n = 3$). **(C)** One-way ANOVA revealed no significant differences between shed DNER in $Bace1^{-/-}$, $Bace2^{-/-}$, and dKO CSF, as compared with WT ($P = 1.00$, $P = 1.00$, and $P = 0.77$, respectively; $n = 3$). **(D)** No significant differences were seen between shed VCAM1 in $Bace1^{-/-}$, $Bace2^{-/-}$, dKO, and WT CSF by one-way ANOVA ($P = 0.76$, $P = 1.00$, and $P = 1.00$, respectively; $n = 3$).

despite its relevance for AD therapy, very little is known about BACE2 expression and function in the central nervous system (CNS). Here, we show that *Bace2* mRNA is present in subsets of neurons, in oligodendrocytes, and in some cells expressing the astrocyte marker *Glast* and lining the lateral ventricle (Fig 1). Their localization in the subventricular zone suggests that they are astrocyte-like neural stem cells (Kokovay et al, 2010), which are known to express VCAM1 (Kokovay et al, 2012). Recent single-cell sequencing shows *Bace2* expression mainly in oligodendrocytes (Zhang et al, 2014), and in subsets of neurons and astrocytes (Zeisel et al, 2015), in agreement with our in situ hybridization experiments. In our study, the highest expression levels of *Bace2* mRNA are seen in oligodendrocytes of the fiber tracts in young mice and in neurons of the ventral hippocampus in adult animals. In comparison, the expression pattern of *Bace1* is much wider with many neurons of the CNS, even those devoid of *Bace2*, displaying *Bace1* mRNA, in line with previous conclusions that it is the predominant β-secretase in the brain (Cai et al, 2001; Vassar et al, 2014; Barão et al, 2016). Given the overlapping expression of *Bace2* with *Bace1* in neurons and oligodendrocytes, it seems logical to speculate on the redundancy of the two enzymes. Both secretases can indeed cleave similar substrates like APP (Dominguez et al, 2005; Bettegazzi et al, 2011), as well as PLXDC2 (Dislich et al, 2015) and FGFR1, which was shown in this study. However, as we will discuss in the following paragraphs, our data also point toward specific processing of VCAM1 (in vitro and in vivo) and DNER (in vitro primary cultures) exclusively by BACE2.

Having established the particular expression pattern of *Bace2* in the brain, we wondered whether there were any specific substrates for BACE2 that could further shed light on its function in the CNS. We focused our attention first on the CSF and decided to compare material from dKO ($Bace1^{-/-}$ and $Bace2^{-/-}$) with single $Bace1^{-/-}$ mice. As shown in Fig S2, the differences observed in transmembrane proteins were small and, actually, the strongest changes were up-regulated proteins without transmembrane domains, making them very unlikely BACE2 substrate candidates. Thus, in contrast to BACE1 deficiency (Dislich et al, 2015), additional depletion of BACE2 did not reveal major BACE2 substrates secreted into the CSF.

We turned to mixed primary glia cultures, where BACE2 was shown to be active (Dominguez et al, 2005). As shown in Fig 2A, we identified four novel BACE2 substrates, namely, VCAM1, FGFR1, DNER, and PLXDC2, the latter two previously reported as BACE1 substrates in other cell types and mouse CSF (Kuhn et al, 2012; Zhou et al, 2012; Stützer et al, 2013; Dislich et al, 2015). VCAM1 and DNER secretion was strongly reduced in $Bace2^{-/-}$ cells and addition of the BACE inhibitor CpJ to WT or $Bace1^{-/-}$ glia inhibited their secretion, indicating that their processing is mainly performed by BACE2 in these cells (Fig 3). The evidence that FGFR1 and PLXDC2 are BACE2 substrates is more circumstantial. As no antibodies were available for FGFR1 and PLXDC2 that detect their expression at the endogenous level, we had to rely on overexpression systems to prove that BACE2, as well as BACE1, can process both (Fig 4).

VCAM1 mediates the adhesion of white blood cells to the vascular endothelium, thereby allowing immune cells to infiltrate through blood vessels (Barreiro et al, 1998). It is also involved in leukocyte homing during atherosclerosis (Gamrekelashvili & Limbourg, 2016). In the brain, VCAM1 is found at high levels in astrocytes (Zhang et al, 2014) and is also expressed by type B neuronal stem cells at the rodent subventricular zone, where it helps maintain the rosette arrangement of these cells and regulates their lineage progression (Kokovay et al, 2012). DNER is a transmembrane protein that mediates Notch signaling through cell–cell interactions in neurons (Eiraku et al, 2002; Saito & Takeshima, 2006). It is involved in the maturation of Bergman glia (Eiraku et al, 2005; Saito & Takeshima, 2006). DNER has previously been identified as a BACE1 substrate in pancreatic β cells (Stützer et al, 2013) and neurons (Kuhn et al, 2012), so it is surprising that in cultured glia cells it is not shed by BACE1, but rather exclusively by BACE2 (Fig 3J and L). According to recent RNAseq databases, *Dner* mRNA is expressed in oligodendrocyte precursor cells and astrocytes, as well as neurons (Zhang et al, 2014; Zeisel et al, 2015), and may have distinct functions in the different cell types.

FGFR1 is involved in signaling pathways of a variety of cells. In the mouse brain, it is expressed at high levels in astrocytes, as well as in

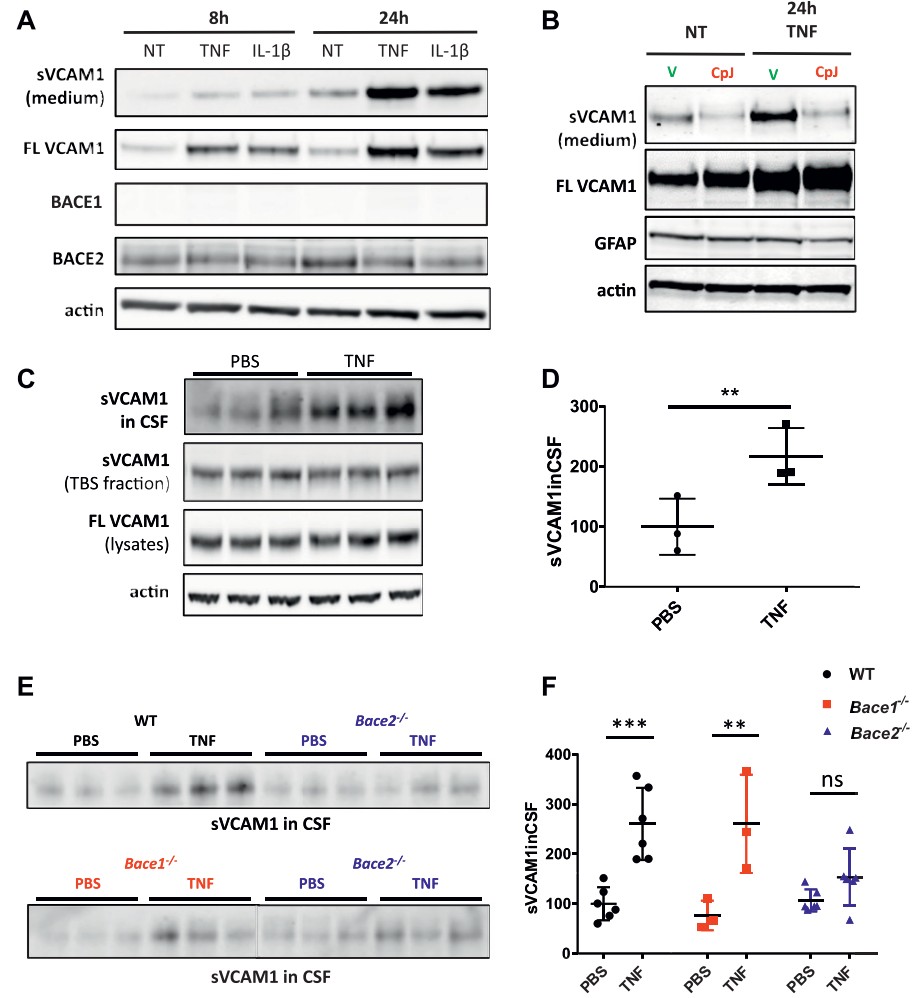

**Figure 6.  Validation of substrate under pro-inflammatory challenge in vitro and in vivo.**
**(A)** VCAM1 in medium and lysates of primary mixed glia culture treated with murine recombinant TNF (10 ng/ml) or IL-1β (10 ng/ml) for 8 h and 24 h (n = 3). Control blots for BACE1, BACE2, and actin are shown. **(B)** VCAM1 in medium and lysates of primary mixed glia cells treated with murine recombinant TNF (10 ng/ml) for 24 h, or treated with vehicle or CpJ inhibitor (10 µM) (n = 3). Control blots for GFAP and actin are shown. **(C)** VCAM1 in triplicates of CSF, TBS fraction, and total cell lysates of cortices from 11-mo-old WT male controls injected with PBS or treated with 250 µg/kg TNF. Actin is used as a loading control. **(D)** VCAM1 shedding into CSF is up-regulated upon TNF treatment (unpaired *t* test, *P* = 0.04; n = 3), whereas no changes are observed in full-length VCAM1 or soluble VCAM1 in TBS fraction shown in (C). **(E)** VCAM1 in triplicates of CSF of 11-mo-old WT or *Bace2*$^{-/-}$ mice treated with saline or 250 µg/kg TNF (top panel) and VCAM1 in triplicates of CSF of 11-mo-old *Bace1*$^{-/-}$ or *Bace2*$^{-/-}$ mice treated with saline or 250 µg/kg TNF (bottom panel). **(F)** Two-way ANOVA reveal significant differences in shed VCAM1 between treated and untreated WT and *Bace1*$^{-/-}$ mice (*P* = 0.0004 and *P* = 0.005, respectively; n = 3), but no significant differences in shed VCAM1 between treated and untreated *Bace*$^{-/-}$ mice (*P* = 0.69).

oligodendrocyte precursor cells (Zhang et al, 2014). PLXDC2 is a transmembrane receptor for the multifunctional pigment epithelium-derived factor important for neural growth, stem cell development, angiogenesis, and cancer cell growth, among other functions (Cheng et al, 2014). PLXDC2 is a mitogen and increases neurogenesis, resulting in thickening of the neural tube at the early stages of development (Miller et al, 2007; Miller-delaney et al, 2011). Interestingly, PLXDC2 was previously identified as a BACE1 substrate in human epithelial cell lines stably expressing BACE1 (Hemming et al, 2009), and its secretion is also affected by BACE1 deletion in mouse CSF (Dislich et al, 2015). In addition, PLXDC2 was identified as a BACE1 substrate in primary mouse neurons in two independent studies (Kuhn et al, 2012; Zhou et al, 2012).

As indicated, we did not see any major changes in shed proteins in the CSF by genetic inactivation of *Bace2* on top of *Bace1* (Fig S2). We revaluated the effect on the secretion of the two major BACE2 substrates identified in the primary cell culture, that is, VCAM1 and DNER, for which, coincidentally, good antibodies were also available. As shown in Figs 5 and S3, we did not see any effects on these two major BACE2 substrates in vivo, in agreement with our unbiased

proteome analysis (Fig S2), suggesting that VCAM1 and DNER are not prominently shed by BACE2 at baseline conditions in vivo in the brain.

Interestingly, the most prominent substrate of BACE2, VCAM1, is known to be up-regulated during inflammation (van Loo et al, 2006). We confirmed this in our *Bace1*$^{-/-}$ mixed glia culture and found that further increase of inflammation in response to TNF or IL-1β additionally increased the amount of VCAM1 secreted into the media (Fig 6A). The increase was inhibited with the BACE inhibitor CpJ (Fig 6B), confirming that additional shedding was indeed BACE2-dependent. Of note, the increase is likely a consequence of increased VCAM1 expression, and not necessarily caused by direct effects on BACE2 expression and/or activity (Fig 6A). We next evaluated whether we could detect BACE2 processing of VCAM1 under inflammatory conditions in vivo. We treated mice for 24 h with TNF, adapting a protocol previously used to induce inflammation in mouse brain (Biesmans et al, 2015). To our surprise, TNF treatment resulted in a strong up-regulation of VCAM1 in the CSF of WT and *Bace1*$^{-/-}$ animals, but did not have an effect on VCAM1 shedding in *Bace2*$^{-/-}$ mice.

This intriguing result not only validates VCAM1 as an exclusive and physiologically relevant BACE2 substrate, but also implicates BACE2 in an inflammatory response pathway. Interestingly, VCAM1 is up-regulated upon injury not only in rat brains (Zhang et al, 2015), but also peripherally in AD patients (Lai et al, 2017) and increases in the CSF with age (Li et al, 2017). This suggests that similar mechanisms of inflammation-induced up-regulation of VCAM1 are relevant in humans. Further work, however, is needed to understand the exact implications of VCAM1 shedding by BACE2 under such conditions. Nonetheless, this work suggests that BACE2 function will only become apparent under specific stress conditions, explaining why until now so little effects at the phenotypic level have been described in BACE2-deficient animals.

An important lesson is to be deduced from these observations for the ongoing BACE1/2 inhibitor clinical trials. Our data predict that side effects caused by BACE2 cross-inhibition in the brain are not going to become apparent under steady-state conditions. However, BACE inhibitors are likely to be given for many years to patients and it is difficult to know what long-term effects chronic suppression of VCAM1 and other BACE2 substrates might cause. One should also take into account that AD is accompanied by CNS and maybe peripheral inflammatory alterations. It might be useful to measure VCAM1 as a marker for inflammation, as well as BACE2 activity, in the CSF and plasma of patients during the clinical trials.

# Materials and Methods

### Animals

The following mouse strains were used in this study: wild-type (WT) C57BL/6J@Rj (Janvier Labs), $Bace1^{-/-}$ (Dominguez et al, 2005), C57BL/6-Bace2<tm1.2> ($Bace2^{-/-}$) (RIKEN Brain Science Institute), and double $Bace1^{-/-}$ $Bace2^{-/-}$ KO (dKO) mice (Dominguez et al, 2005). The dKO mouse line was produced by breeding the aforementioned $Bace1^{-/-}$ mice and $Bace2\Delta E6$ mice, which harbor a deletion of exon 6 of $Bace2$ that contains the active site of the enzyme. The RIKEN $Bace2^{-/-}$ mice used in this study lack the BACE2 protein completely. All the strains are on a C57BL/6 background and were maintained on a 12/12-h light–dark cycle with ad libitum food and water. All experiments were approved by the ethics committee of the University of Leuven and performed in accordance with the Belgian and European Union regulations.

### Multiplex fluorescent in situ hybridization (RNAscope)

In situ hybridization was performed using the RNAscope Fluorescent Multiplex Assay (ACD Bio) according to the manufacturer's instructions. Briefly, whole hemispheres of WT adult (4 mo) or WT young (P16–P20) mouse brains were frozen in molds filled with optimum cutting temperature (OCT) embedding matrix (Tissue-Tek). Then, 14- to 16-μm sections were prepared on Superfrost Plus slides (Thermo Fisher Scientific) using the NX70 cryostat (Thermo Fisher Scientific). The sections were fixed in 4% PFA and dehydrated using a series of ethanol dilution steps. Protease digestion was carried out for 20 min at RT using Pretreat 4 for fresh frozen tissue

provided in the RNAscope kit. Hybridization proceeded for 2 h at 40°C. The following probes were used: $mBace1$-C2, $mBace2$-C1, $mSyp$-C3 (neuronal marker), $mGlast$-C3 (astrocyte marker), and $mMbp$-C3 (oligodendrocyte marker). Brain-specific housekeeping genes $mPolr2a$, $mPpib$, and $mUbc$ were used as high-, medium-, and low-expressing positive control probes, respectively. Bacterial $DapB$ probe was used as a negative control. Probe detection was performed using the four amplification reagents provided in the RNAscope kit. Images were acquired using the Leica SP8× confocal microscope and analyzed using the ImageJ software.

### Mouse CSF collection

CSF was extracted from WT, $Bace1^{-/-}$, $Bace2^{-/-}$, and dKO mice according to a previously described protocol (Liu & Duff, 2008) with minor adjustments. A mixture of ketamine (100 mg/kg) and xylazine (10 mg/kg) in PBS was administered intraperitoneally to anesthetize the mice. The animal was placed on a heating pad and secured in the stereotaxic instrument with the head of the animal held at a 135° angle. Hair on the head of the mouse was moved aside and held by Duratears cream and the skin of the head and neck overlying the cisterna magna was cut. Muscles and subcutaneous tissue were gently pulled aside and secured using retractors to reveal the triangular cisterna magna. The surgical field was cleaned from any blood, wiped three times with PBS, and dried if necessary. The dura was punctured with a pulled glass capillary (Sutter Instrument) attached to a syringe by tubing. The CSF was collected from the cisterna magna by applying negative pressure and transferred into a 0.5-ml LoBind tube (Eppendorf). Following a centrifugation step at 1,500 $g$ for 10 min, clean CSF was transferred to a fresh tube and frozen at –80°C until analysis.

### Primary glia culture and Bace inhibition experiments

All cell culture media, PBS, HBSS, trypsin, and supplements were purchased from Invitrogen. Glia cultures were prepared from P3 mouse pups. Briefly, the cortices were cleared from meninges, cut into small pieces, digested by trypsin, and filtered through a 70-μm filter, and cells from two pups were plated on 6-cm uncoated petri dishes. Glia cultures were maintained in MEM with Earle's salt and L-glutamine, 12.5% FBS, 0.6% glucose, and pen-strep. Once confluent, each 6-cm dish was frozen in 1 ml freezing medium (90% FBS and 10% DMSO) to establish a frozen stock of glia. For Bace inhibition, a previously characterized small molecule CpJ was used, which is an aminodihydrothiazine derivative inhibiting both BACE1 and BACE2 secretases (Kobayashi et al, 2007; Stützer et al, 2013).

### Sample preparation for LC–MS/MS analysis

CSF samples (6 $Bace1^{-/-}$, 5 dKO) were processed as previously described (Pigoni et al, 2016). Briefly, 5 μl CSF was alkylated using dithiothreitol (Biomol) and iodoacetamide (Sigma-Aldrich). Proteolytic digestion was performed by using 0.1 μg LysC (Promega) for 4 h and subsequently 0.1 μg trypsin (Promega) for 16 h at RT in the presence of 0.1% (w/v) sodium deoxycholate (Sigma-Aldrich). Deoxycholate was removed after acidification by centrifugation (16,000 $g$, for 10 min at 4°C). Peptides were purified and desalted

using C18 stop and go extraction (Rappsilber et al, 2003). For LC–MS/MS analysis, the peptides were dissolved in 20 μl 0.1% formic acid (FA).

For proteomics experiments, eight vials of frozen stock *Bace1*<sup>−/−</sup> glia (representing 16 animals) were plated in 10 × 10-cm petri dishes (Fig S5A). All dishes were labeled with Click-IT ManNAz Metabolic Glycoprotein Labeling Reagent (Life Technologies). Five dishes were treated with 10 μM CpJ, whereas the remaining five were treated with 20% Captisol vehicle as control, for 48 h in 5% FBS-containing medium. Conditioned medium was collected, filtered through a 0.45-mm PVDF filter (Millex), and concentrated using 30-kD Amicon Ultracel Centrifugal filters (Millipore). The proteomics experiment was repeated three times using independent cultures each time. The secretome protein enrichment with click sugars was used for sample preparation of conditioned cell culture medium from vehicle and CpJ-treated glial cells as previously described (Kuhn et al, 2012). This protocol facilitates the enrichment of metabolic-labeled glycoproteins in the presence of serum proteins. Briefly, azide-labeled protein glycans were biotinylated with dibenzylcyclooctyne–PEG4–biotin (DBCO–PEG4–biotin) conjugate (Jena Bioscience) using click chemistry. The biotinylated glyco-proteins were pulled down using high-capacity streptavidin aga-rose (Thermo Fisher Scientific). The proteins were eluted from the beads in 150 μl Laemmli buffer supplemented with 8 M urea and 3 mM biotin at 95°C for 5 min. Proteins were separated by SDS–PAGE (10%). Coomassie staining was used to visualize the proteins. Each lane was cut into 14 fractions excluding the albumin band at 60 kD. The gel slices were subjected to protein in-gel digestion using 150 ng of trypsin (Promega) per slice (Shevchenko et al, 2007). After peptide extraction, the samples were dried using vacuum centri-fugation. For LC–MS/MS analysis, the peptides were dissolved in 20 μl 0.1% FA.

### LC–MS/MS analysis

An Easy nLC-1000 (Thermo Fisher Scientific) equipped with a col-umn oven (Sonation) online coupled via a nano-electrospray ion source (Thermo Fisher Scientific) to either a Q-Exactive or a Velos Pro Orbitrap Mass Spectrometer (Thermo Fisher Scientific) was used for LC–MS/MS analysis. Peptides were separated on a self-packed C18 column (300 mm × 75 μm, ReproSil-Pur 120 C18-AQ, 1.9 μm, Dr. Maisch GmbH HPLC) using a binary gradient of water (A) and acetonitrile (B) supplemented with 0.1% FA (CSF samples: 0 min, 2% B; 3:30 min, 5% B; 137:30 min, 25% B; 168:30 min, 35% B; 182:30 min, 60% B; 185 min, 95% B; 200 min, 95% B; Gel fractions of glial secretome: 0 min, 2% B; 3:30 min, 5% B; 48:30 min, 25% B; 59:30 min, 35% B; 64:30 min, 60% B).

CSF samples were analyzed on a Q-Exactive system using a resolution of 70,000 for full mass spectrometry spectra. The 10 most intense peptide ions per spectrum were chosen for frag-mentation using higher energy collisional dissociation (AGC target: 1E+5; NCE: 25%). A dynamic exclusion of 120 s was applied for fragment ion spectra acquisition.

Full mass spectrometry spectra were acquired at a resolution of 60,000 (AGC target: 3E+6). The 10 most intense peptide ions per spectrum were chosen for collision-induced dissociation within the ion trap (AGC target: 1E+4; NCE: 35%; and activation Q: 0.25).

A dynamic exclusion of 90 s was applied for fragment ion spectra acquisition.

### LC–MS/MS data analysis

Database search and LFQ were performed with the software MaxQuant (version 1.5.4.1, maxquant.org) using default settings (Cox et al, 2014). A canonical database of the reviewed reference mouse proteome (UniProt, download: June 8, 2016; 16,798 entries) was used for database search. The false discovery rate for both peptides and proteins was adjusted to <1% using a target and decoy approach (concatenated forward/reverse database). The "match between runs" option was enabled using a time window of 1.5 min. The LFQ intensities were calculated based on unique peptides requiring at least two ratio counts.

For relative quantification of CSF samples, LFQ intensities were log2 transformed and the average log2 ratio was calculated. At least three quantification values per group were required for statistical analysis. A Student's *t* test was applied to check for significant changes in protein abundance.

For relative quantification of the glial secretome, the LFQ in-tensity ratios of CpJ and vehicle-treated samples were calculated separately for each experiment. At least three LFQ ratios were required for statistical analysis. All LFQ ratios were log2 trans-formed and a one-sample *t* test ($\mu_0$ = 0) was applied to check if the average log2 LFQ ratio is different from zero.

For further data analysis, protein accessions were matched against the subcellular location database of UniProt (membrane [SL-0162]; cytoplasm [SL-0086]; mitochondrion [SL-0173]; secreted [SL-0243]; nucleus [SL-0191]; cell membrane [SL-0039]; GPI-anchor [SL-9902]; single-pass type I membrane protein [SL-9905]; single-pass type II membrane protein [SL-9906]; single-pass type III membrane protein [SL-9907]; single-pass type IV membrane protein [SL-9908]; and multi-pass membrane protein [SL-9909]). In addi-tion, all proteins were matched against the UniProt keyword da-tabase for glycoproteins [KW-0325].

### Validation experiments in primary glia

Glia from *Bace1*<sup>−/−</sup> and *Bace2*<sup>−/−</sup> pups and their respective WT lit-termates were cultured simultaneously. For each genotype, one vial (representing two animals) was plated into a 10-cm dish (Fig S5B). At confluency, fresh serum-free medium was applied and collected after 24 h. This experiment was repeated three times with in-dependent cultures. In addition, WT, *Bace1*<sup>−/−</sup>, and *Bace2*<sup>/−</sup> glia were cultured in two 10-cm dishes each until confluency and treated with CpJ or vehicle for 24 h in serum-free medium (Fig S5C). Again, two animals per genotype per treatment were used and all experiments were repeated at least three times. All collected media were concentrated using the 30-kD Amicon Ultracel Centrifugal filters. Cell lysates were prepared in 50 mM Tris-HCl, pH 7.4, 150 mM NaCl, 2 mM EDTA, and 1% Triton X-100 supplemented with complete protease inhibitor (Roche Applied Science). Cells were lysed for 20 min on ice and cleared by centrifugation at 14,000 *g* for 15 min. Protein concentrations were measured using standard BCA assay (Pierce).

## Western blotting

20 µg of total protein was separated by SDS–PAGE on 4–12% BisTris gels and transferred onto nitrocellulose membranes. The membranes were blocked in 5% milk prepared in TBS-0.1% Tween and incubated in primary antibodies overnight at 4°C. The following primary antibodies were used: rabbit monoclonal anti-BACE1 (Cell Signaling, #5606, clone D10E5), rabbit polyclonal anti-BACE2 produced in-house, goat polyclonal anti-VCAM1 IgG (Thermo Fisher Scientific, PA5-47029), goat polyclonal anti-DNER IgG (R&D Systems, AF2254), mouse monoclonal anti-FGFR1 alpha antibody [M2F12] (Abcam, ab829), mouse monoclonal anti-FLAG M2 (Sigma-Aldrich, F3165), rabbit polyclonal anti-HA antibody (Clontech, 631207), and mouse monoclonal anti-β-actin (Sigma-Aldrich, A5441). The following secondary antibodies were used: goat polyclonal anti-mouse IgG-HRP conjugate (Biorad, 170-6516), goat polyclonal anti-rabbit IgG-HRP conjugate (Biorad, 170-6515), rabbit polyclonal anti-goat IgG-HRP conjugate (DAKO, P0449). Blots were developed using the ImageQuant LAS 4000 mini machine (GE Healthcare) and band intensities were quantified with AIDA image analyzer software (Raytest). The levels of the full-length (FL) proteins were normalized to actin levels.

## COS-1 cell culture and transfections

COS-1 cells were seeded in 10% FBS supplemented DMEM/F-12 cell culture medium in six-well plates. All cell culture medium and supplements were purchased from Invitrogen. Transfections were performed using the TransIT-LT1 Transfection Reagent (Mirus Bio LLC) according to the manufacturer's instructions. The following constructs were used: mouse *Bace1* and *Bace2* in pSG5** vector as in Dominguez et al (2005), except that *Bace2* construct also contained a C-terminal FLAG-tag for facilitated detection. Mouse *Plxdc2* in pcDNA3.1 vector, tagged with HA-tag at the N-terminus and FLAG at the C-terminus, was previously used (Dislich et al, 2015). *FGFR1* plasmid in pWZL vector (Boehm et al, 2007) was purchased from AddGene and re-cloned into the pSG5 vector by Gibson assembly. Empty pcDNA3.1 vector was used as control in all overexpression experiments. For validation of PLXDC2 and FGFR1 as BACE2 substrates, the substrate constructs were transfected either alone or together with either *Bace1* or *Bace2* constructs. Co-transfected wells were treated with either 10 µM CpJ or vehicle; an empty vector transfection was performed as control. Three independent transfection experiments were performed for each substrate.

## Mouse brain tissue collection and sample prep

Animals were euthanized using $CO_2$ and decapitated. The brain was subdissected, snap-frozen in liquid nitrogen, and stored at −80°C until further analysis. Homogenates were prepared from cortices of one hemisphere in 300 µl TBS buffer (50 mM Tris–HCl, pH 7.6, and 150 mM NaCl) supplemented with complete protease inhibitor (Roche Applied Science) using a Teflon glass homogenizer. Following centrifugation at 14,000 $g$ for 15 min, the supernatants were placed in a fresh tube and ultracentrifuged at 70,000 $g$ for 30 min yielding the soluble TBS fractions. The remaining cell pellets were lysed in 50 mM Tris-HCl, pH 7.4, 150 mM NaCl, 2 mM EDTA, and 1%

Triton X-100 supplemented with complete protease inhibitor (Roche Applied Science) for 20 min on ice and cleared by centrifugation at 14,000 $g$ for 15 min. Protein concentrations were measured using standard BCA assay (Pierce) and the samples were analyzed by Western blotting as described above to detect the shed soluble fragments in TBS fraction and FL proteins in the total cell lysates.

## TNF treatment of glia cultures and mice

Primary glia cultures were prepared from *Bace1*$^{−/−}$ mice as described above and two vials of frozen stock (representing four animals) were seeded into six-well plates. The cells were treated with recombinant murine TNF (10 ng/ml) or IL-1β (10 ng/ml) for 8 h or 24 h. Three repeats of independent cultures were performed. For inhibition experiments, the cells were pretreated with 10 µM CpJ or vehicle overnight. The next morning, the cells were simultaneously treated with 10 µM CpJ or vehicle in the presence of 10 ng/ml TNF or CpJ or vehicle alone. Three repeats of independent cultures were performed. Medium was processed and total cell lysates were prepared as described above.

For in vivo experiments, WT, *Bace1*$^{−/−}$, and *Bace2*$^{−/−}$ mice were treated with PBS saline or 250 µg/kg TNF for 24 h; afterward CSF was extracted as described above, the mice were euthanized, and the brain was dissected and kept for analysis.

## Statistical analysis

Statistical analysis for Western blot experiments was performed using the GraphPad Prism Software. For validation experiments in glia cells, immunoblot band intensity was normalized to that of a vehicle-treated sample of each independent experiment and paired $t$ test was used to determine statistical differences. Analysis of proteins in the CSF, TBS fraction, and total cell lysates of brain homogenates from different genotypes was performed by one-way ANOVA with Bonferroni post hoc test. VCAM1 measurement in the CSF of WT mice treated with PBS or TNF was analyzed using unpaired $t$ test. VCAM1 in CSF from mice of different genotypes treated with PBS or TNF was analyzed using two-way ANOVA with Tukey's multiple comparisons test. All Western blot quantifications are presented as mean ± SD.

# Supplementary Information

# Acknowledgements

This work was funded by the Agency for Innovation by Science and Technology (IWT), the Stichting Alzheimer Onderzoek (SAO), the Fund for Scientific Research, Flanders, the KU Leuven, a Methusalem grant from the KU Leuven, the Flemish Government, and Vlaams Initiatief voor Netwerken voor Dementie Onderzoek (VIND, Strategic Basic Research Grant 135043), the Deutsche Forschungsgemeinschaft (FG2290), the Helmholtz-Israel program,

the centers of excellence in neurodegeneration, and the Breuer Foundation Research award. B De Strooper is supported by the Arthur Bax and Anna Vanluffelen chair for Alzheimer's disease and "Opening the Future" of the Leuven Universiteit Fonds (LUF).

## Author Contributions

I Voytyuk: Data curation, formal analysis, validation, investigation, visualization, methodology, and writing—original draft, review, and editing.

SA Mueller: Data curation, software, formal analysis, investigation, visualization, methodology, and writing—review and editing.

J Herber: Data curation, investigation, methodology, and writing—review and editing.

A Snellinx: Methodology and writing—review and editing.

D Moechars: Conceptualization, resources, supervision, funding acquisition, project administration, and writing—review and editing.

G van Loo: Conceptualization, supervision, and writing—review and editing.

SF Lichtenthaler: Conceptualization, resources, supervision, funding acquisition, and writing—review and editing.

B De Strooper: Conceptualization, resources, supervision, funding acquisition, and writing—original draft, project administration, writing—review and editing.

## Conflict of Interest Statement

B De Strooper was a consultant for Janssen Pharmaceutica and reMYND NV; I Voytyuk, SA Mueller, J Herber, A Snellinx, G van Loo, and SF Lichtenthaler declare no conflict of interest. D Moechars is a researcher of Janssen Pharmaceutica.

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
