## [Reviewer comments · Life Science Alliance]

Bace2 distribution in major brain cell types and identification of novel substrates.

Iryna Voytyuk, Stephan A. Mueller, Julia Herber, An Snellinx, Dieder Moechars, Geert van Loo, Stefan F. Lichtenthaler, Bart De Strooper

DOI: 10.26508/lisa.2017000026

Review timeline:

Submission Date:	29 January 2018
Revision Received:	29 January 2018
Editorial Decision:	30 January 2018
Accepted:	01 February 2018

Report:

(Note: Letters and reports are not edited. The original formatting of letters and referee reports may not be reflected in this compilation.)

Please note that the manuscript was previously reviewed at another journal and the reports were taken into account in inviting a revision for publication at *Life Science Alliance* prior to submission to *Life Science Alliance*.

1st Revision – authors' response

29 January 2018

Referee #1 (Remarks for Author):

Manuscript "Bace2 distribution in major brain cell types and identification of novel substrates" by I. Voytyuk et al., describes the distribution of β -site APP-cleaving enzyme 2 (Bace2) in mouse brain and the effects of Bace inhibition using a multidisciplinary approach (e.g., in situ hybridization, immunoblot analysis, proteomic analyses) in relevant animal and cellular models. Four new substrates of Bace2 were also described. Bace2 expression has been well-documented in the periphery, but distribution in the brain remains largely unknown. There is heightened interest in Bace2 expression levels in neurons and glia due to the development of Bace1 inhibitory compounds for the potential treatment of Alzheimer's disease. While there is strength in the molecular and proteomic approaches used in the study, there are several experimental concerns that weaken the analysis and the interpretations of the results.

1. Potentially underpowered studies.

The major critique of the entire work is that the number of animals and cell cultures is not given for the majority of the experiments. The Reviewer found it very confusing, considering the number of experiments and models being used that a breakdown of the number of mice per genotype, per age, per experiment and cell culture replicates was not given in the Materials and Methods. Scant usage of animals/cells was given in some of the Figure Legends (e.g., Figure 3), which was more confusing than helpful. The authors are urged to create a flow diagram or table for animal & cell culture usage for the entire series of experiments with the numbers used for statistical analysis.

The Material and Methods sections have been updated with relevant information on the number of repeats and specifics of primary cell culture.

Flow chart diagrams have been added in Supplementary Figure 3.

2. Statistical analyses equivocal.

A second major issue in this manuscript is statistical methods are not described in detail and the actual analyses appear equivocal for many of the experiments. It is also unclear that enough samples/conditions/cell culture experiments were run to actually conduct appropriate statistical assessments. For example, several comparisons (Figures 3D, 3E, 3F, 3J, 3K, 3L, 4A, 4B among others): or "representative" 2-3 experiments with no formal statistical testing. Especially given the

observation that even large differences may not be significant (e.g., Fig. 3C) the validity of drawing such conclusions has not been demonstrated. Statistical analyses are either incomplete or not presented for in situ hybridization and immunoblot analyses, and the proteomic analyses were not well described. Accordingly, many of the findings should be considered preliminary unless appropriate statistical muster can be demonstrated.

Statistical analysis was never performed on cases with $n < 3$ in the original draft, however, additional experiments have now been added to provide $n = 3$ for all experiments.

Referee #2 (Remarks for Author):

Bace1 (β -site APP-cleaving enzyme 1) inhibitors are clinically used for Alzheimer's disease (AD) treatment, however these drugs can also inactivate the homologue enzyme Bace2. The role of Bace2 in the brain is not well-characterized and the possible side effects derived from cross-inhibition of Bace2 are unknown. In the current manuscript, Voytyuk and colleagues have determined the localized Bace2 mRNA expression in the brain and have also identified novel substrates for Bace2 in glial cultures. Interestingly, the shedding of VCAM1, one of these novel substrates, is only observed in vivo under pro-inflammatory conditions. Since inflammatory events accompanies AD, Voytyuk's data suggests that VCAM1 cleavage might be an off-target effect of the Bace1/2 inhibitors during the treatment of AD patients.

The manuscript is well-written and the data is in general clearly presented, however the biological relevance of the findings is not analyzed. In my opinion, the manuscript is merely descriptive and lacks a deeper examination on the possible effects derived from the inflammation-induced VCAM1 shedding by Bace2.

SPECIFIC POINTS

1. The end point of the manuscript is to show that VCAM1 is shed in pro-inflammatory conditions and that the cleavage depends on Bace2. However, the evaluation of biological significance of the results is very poor. Which is the function of sVCAM1 in AD? Is Bace2 activity important for the neuroinflammatory process?

The function of Vcam1 or Bace2 in AD or inflammation is not known. In the manuscript we discuss the literature showing upregulation of Vcam1 during aging and various stressful conditions. As we discuss in the manuscript, Vcam1 is upregulated after injury in rat brains (Zhang et al, 2015), increases in the CSF with age (Li et al, 2017) and peripherally in AD patients (Lai et al, 2017). However, the mechanism of the upregulation and the consequences are not yet studied, although they seem to be conserved through species. Data from our experiments would support that Bace2 activity (at least through Vcam1 cleavage) is important. However, the function of Bace2 and/or Vcam1 under inflammatory conditions would not fall under the scope of the current study, which is mainly focused on identifying Bace2 expression pattern in the brain and its substrates. It would be, nonetheless, an interesting follow-up aiming to understand the exact implications of VCAM1 shedding by BACE2 under inflammatory conditions in humans.

2. How is Bace2 activity regulated by inflammatory stimuli? The expression of Bace2 is not altered by TNF α stimulation, instead the amount of FL VCAM1 is importantly increased in the glial cultures. Is it possible that what the authors observe is actually an overexpression of total VCAM1 and therefore an increase of the soluble fraction of the protein?

We did not determine whether Bace2 is activated by inflammatory stimulus. We do observe that its expression levels remains constant, and so in the manuscript we postulate that the increased expression of Vcam1 causes an excess of Vcam1 in the cell, and this excess is cleaved by Bace2. So Bace2 likely takes over when too much Vcam1 is present. Because this excess is physiologically relevant (as discussed in the point above), the role of Bace2 might be important.

3. VCAM1 can be cleaved by other enzymes, such as ADAM17 (Garton et al, 2003). How specific is Compound J?

Compound J is a specific inhibitor of Bace1 and Bace2. We performed an experiment with a metalloprotease inhibitor (GM6001) that indeed proves that Vcam1 is partially cleaved by Bace2, and partially by metalloproteases, likely Adam17, as seen in the study of Garton et al, 2003 in JBC.

4. The authors could not validate the cleavage of Dner in vivo. Is this also depending on an inflammatory context?

We have performed experiments with Dner in primary glia culture (see Supplementary Figure 4 added in the updated version of the manuscript). Surprisingly, upon treatment with pro-inflammatory agents TNF or IL1- β Dner expression in the primary glia culture is markedly decreased, suggesting an opposite effect to that on Vcam1. Although this is very interesting, it poses technical difficulties, as shown, for detection of Dner in the concentrated medium of the glia cells. The basal levels of Dner shedding is already low and when inhibited with CpJ, it is barely detectable. Treatment with pro-inflammatory agents will move the levels of shed Dner into undetectable ranges, as already seen in Supplementary Figure 4. Furthermore, to perform these experiments in vivo, the levels of shed Dner into the CSF would be undetectable as well, as detection did not work in basal conditions.

5. Why are two different Bace2 KO models used? Why was the Bace2 $\Delta E6$ mutant used in combination with Bace1 $^{-/-}$ for the dKO? The authors justify that the changes in protein expression observed in the dKO versus Bace1 $^{-/-}$ are probably related to the absence of Bace2 in the dKO but this is not a plausible explanation since Bace2 $\Delta E6$ lacks only the active domain.

Our homemade Bace2 $\Delta E6$ harbor a deletion of exon 6, which contains the active site of the enzyme. This results in a 4.5 kb deletion, which could still produce a Bace2 protein (unless there is nonsense-mediated RNA decay) (Dominguez et al, 2005). At the time this mouse line was generated, it was impossible to examine Bace2 expression on the protein level, with no commercially available antibodies specifically detecting the protein. Because we wanted to use a Bace2 KO as control for testing our homemade antibodies, we ordered a new full Bace2 KO mouse line from Riken (No.RBRC02633). Since then we have purified and tested homemade produced antibodies for Bace2, using the Bace2-R KO mouse line as control. We have purified one antibody that specifically detects Bace2 in primary mouse glia and brain homogenates (a band seen at ~55 kDa that disappears in the Bace2-R KO lane), however, it has multiple unspecific background bands (see image below). Therefore, when tested on dKO mice (cross between our homemade Bace1 $^{-/-}$ and Bace2 $\Delta E6$), the Bace2 band at ~55 kDa is no longer seen and the mouse appears as a KO for Bace2. Nonetheless, due to presence of multiple background band we cannot exclude that Bace2 produced by dKO and Bace2 $\Delta E6$ mice runs at a lower molecular weight.

6. Figure 2a shows Bace2 expression instead of activity as it is mentioned in the text.

This is now corrected in the manuscript.

7. In Figure 2b, most of the GFAP+ cells are also O4+. Could you explain this result?

O4 has been commonly used as the earliest recognized marker specific for the oligodendroglial lineage. However, it is a marker of progenitor cell that have not acquired yet a defined identity. Astrocytes and oligodendrocytes both develop from glial progenitors in the brain. Maybe it is possible that low levels of this marker still remain in the cells that committed to the astrocytic lineage, and these levels essentially disappear once the astrocyte is mature. Cells in culture, glia especially, differ in their expression profiles from their counterparts in vivo. Therefore, multiple markers need to be used in order to establish exact cell type identity. Because it created confusion we have removed, the immunostaining figure panel from the updated version of the manuscript.

8. The authors state that Bace1 levels are constant between the different genotypes but this is not reflected in Fig3D.

Being primary cultures, completely equal levels of any protein is difficult to achieve and some variability between independent cultures is to be expected. I was trying to explain that Bace1 levels fluctuate less than Bace2, which was very variable in this type of culture.

9. The discussion part is mainly a repetition of the results section. This section needs improvement. Also, the authors should be more careful about the expression data exposed: FGFR1 is not mostly expressed in OPCs according to the paper they quote and Dner is also highly expressed in astrocytes (otherwise it would be difficult to explain their data on glial cultures).

The discussion places the findings into context of previously published literature about the identified substrates. The wording has been adjusted to be more precise about the expression levels of the different substrates as interpreted from RNA-seq databases from the labs of Ben Barres and Sten Linnarson.

10. The methodological details on the Compound J are missing.

The compound is previously described and used as an inhibitor of both Bace1 and Bace2 in other studies. Its patent as well as usage is referenced in the manuscript. Concentrations and times of treatments are specified for each experiment.

Referee #3 (Comments on Novelty/Model System for Author):

The authors first demonstrate differential expression of BACE2 in mouse brain with respect to brain regions and cell type using in situ RNA hybridization. The authors then search for BACE2 substrates by comparing proteins in CSF harvested from BACE1/2 KO mice and BACE1 KO mice and identify candidate BACE2 substrates by mass spec. A total of 576 proteins were identified. Of these, 60 proteins were chosen based on statistical analysis and of these, all but four proteins remained after choosing only those having a single transmembrane domain (a likely requirement for acting as a BACE substrate). The authors note that proteins in CSF are derived from large numbers and kinds of brain cells that differentially express BACE2. Due, perhaps, to the high variability encountered and the partial effects on substrate shedding, the authors chose primary mixed glial cell cultures to examine BACE2 shedding of substrate proteins because they had previously shown that these cells express BACE2 activity and presumably they reflect the cell types observed in the histological data.

The authors derived the mixed glial cultures from BACE1 KO mice to focus mainly on BACE2 shedding. They then compared conditioned media from cultures that had been exposed to vehicle or a non-selective BACE inhibitor drug, Compound J. The conditioned media were enriched in glycosylated membrane proteins using the SPECS method (the authors note that 90% of the resulting proteins are single-span membrane proteins -potential BACE2 substrates). Of 246 proteins identified by mass spec four (Vcam1, Dner, Fgfr1 and Plxdc2) were decreased by more than 30% (shed protein fragments). Of these four, Vcam1 and Dner were decreased by 70-80% suggesting that BACE2 is the main protease responsible for their processing. The two others were decreased by 35-48%, suggesting that their shedding was due to BACE2 and other proteases. The authors first validated Vcam1 and Dner in primary mixed glial cultures by showing that WT and BACE1 KO cells shed both proteins but that shedding was strongly decreased in BACE2 KO cells and in cells treated with Compound J. The full-length parent proteins of each did not show significant increases in levels, making it unlikely that shedding was affected by protein expression levels.

The authors were not able to resolve endogenous levels of Plxdc2 and FGFR1 in western blots. They therefore used over-expression in Cos cells as their assay. This suggested that both proteins are cut by BACE2 and BACE1 and perhaps other proteases.

REVIEWER'S CRIT 1: The ability of this system to yield meaningful results is compromised in the opinion of this reviewer because protein over-expression could result in recruitment of alternative proteases that may not be physiologically relevant.

We discuss this limitations of the overexpression system in the manuscript. We do not suggest overexpression as a definite proof of substrate cleavage in physiological context. However, we resort to this approach (previously widely used for validation of Bace1 substrates, among others) due to lack of specific antibodies to detect Fgfr1 and Plxdc2. Vcam1 and Dner were validated in primary cultures, as well as in mouse samples, because antibodies were available.

The authors then sought to validate (as BACE2 targets) Dner and Vcam1 in mouse brain by comparing CFS for endogenous levels of Dner and Vcam1 from WT, BACE double KO, BACE1 KO and BACE2 KO mice. In contrast to the results from primary mixed glial cultures, no differences in shed Dner or Vcam1 was observed. The full-length proteins levels were also not affected. A further test was performed where Compound J was administered to BACE1 KO mice to eliminate all BACE1/2 cleavages. Still, no differences were detected in shedding of either protein suggesting that neither of these proteins is processed by BACE1/2 in the murine brain.

The authors state that because injuring to astrocytes has been shown to result in astrocyte activation, cell culture conditions might replicate cell injury, which might induce a subsequent inflammatory phenotype. The authors also point out that Vcam1 had been shown to be upregulated in another study. This motivated the authors to test shedding of Vcam1 in glial cultures exposed to the proinflammatory cytokines, TNF- α and IL2-b. These experiments showed that shedding of Vcam1 and Dner was increased by these factors in primary cultures by BACE2 processing. The authors then tested adding TNF- α to mice (WT, BACE1 KO, BACE2 KO) and showed that mice treated with TNF- α had a 2.2-fold increase in shedding of Vcam1 in CFS and an increase in shedding in BACE1 KO mice but Vcam1 shedding in BACE2 KO mice was unaffected, concluding

that BACE2 is responsible for increased Vcam1 shedding under proinflammatory conditions.

REVIEWER'S CRIT 2: It is not clear why the same experiments were not performed for Dner. Also, although TNF- α is expected to stimulate an inflammatory response, no evidence of inflammation is provided.

We have performed in vitro experiments for Dner, specifically treatment with TNF and IL1- β . As already commented above, this resulted in downregulation of Dner, making it very difficult to detect the secreted form in the concentrated medium of glia cultures. Because detection of Dner in mouse CSF is already at the border of detectability, we did not see fit to subject more animals to inflammatory treatment.

For in vitro experiments, we performed a Western blot for Gfap, as seen in figure 6B. However, no increase of Gfap at the protein level was detected. It is likely that due to already high inflammatory profile of cultured glia, resulting in high Gfap expression, additional stimulation did not cause further increase. In vivo, we used a previously tested dose of TNF, shown to cause inflammatory response in astrocytes (Biesmans et al 2015). As a control of the TNF injection causing inflammation, we used obvious sickness behavior of the mice, as has also been described by Biesmans et al to occur concomitantly with activation of astrocytes in the CNS.

REVIEWER'S CRIT 3: It would have been quite interesting if the authors had repeated the BACE2 localization/expression experiments (Fig. 1) in mice that had been administered TNF- α . This might have shed some light on why BACE2 processing of Vcam1 was dependent on inflammatory conditions.

We have recently repeated Bace2 localization/expression experiments in mice that had been administered TNF. There were no clear differences detected between mRNA expression in 4 month old mouse brains and brains of TNF treated mice. We performed TNF challenge experiments in vivo at only one time point (24h), and to establish at what time point transcriptional changes may occur, we would need to set up a time course experiment. This would involve the use of many additional mice and we feel that the potential additional information from such experiment is not high enough.

1st Editorial Decision

30 January 2018

Thank you for submitting your revised manuscript entitled "Bace2 distribution in major brain cell types and identification of novel substrates" to Life Science Alliance.

The manuscript was previously reviewed at a different journal and the referee reports have been transferred to Life Science Alliance. You provided a revised manuscript and a detailed point-by-point response to the reports obtained during peer-review elsewhere. During our pre-transfer discussion, I had already outlined that we would expect you to address the previously raised concerns by including the number of experiments conducted / mice used, by providing appropriate statistical tests / display of individual data points, and by text changes. I see that you addressed all these concerns, and I appreciate your detailed response to the concerns raised by the referees and the additional data provided. I am thus happy to accept your manuscript in principle for publication in Life Science Alliance.

Before sending you the official acceptance letter, I would like to ask you to log once more into our system to update a few items and to provide an electronic license form.

Thank you for this very nice contribution to Life Science Alliance and for your enthusiasm for our new open-access journal! I am very much looking forward to publishing your paper.